# 4RC: 4D Reconstruction via Conditional Querying Anytime and Anywhere

**Yihang Luo** [1] **Shangchen Zhou** [1] **Yushi Lan** [2] **Xingang Pan** [1][3] **Chen Change Loy** [1][3]

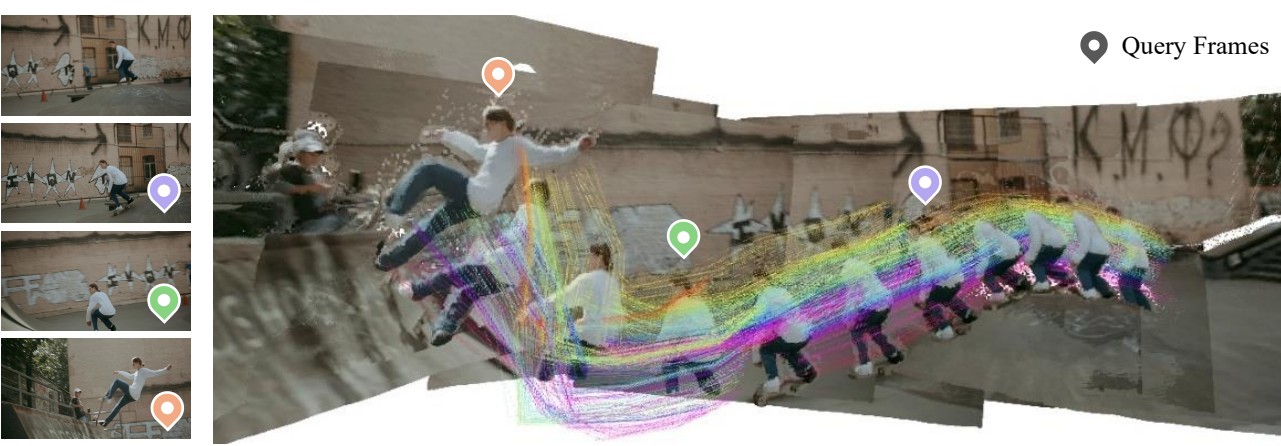

*Figure 1.* **4RC** (pronounced "ARC") enables unified and complete **4D R**econstruction via **C**onditional querying from monocular videos in a single feed-forward pass. It jointly recovers camera poses and dense per-frame geometry, while supporting flexible querying of dense 3D motion from arbitrary source frames to any target timestamp.

## Abstract

We present 4RC, a unified feed-forward framework for 4D reconstruction from monocular videos. Unlike existing methods that typically decouple motion from geometry or produce limited 4D attributes, such as sparse trajectories or two-view scene flow, 4RC learns a holistic 4D representation that jointly captures dense scene geometry and motion dynamics. At its core, 4RC introduces a novel *encode-once, query-anywhere and anytime* paradigm: a transformer backbone encodes the entire video into a compact spatio-temporal latent space, from which a conditional decoder can efficiently query 3D geometry and motion for *any* query frame at *any* target timestamp. To facilitate learning, we represent per-view 4D attributes in a minimally factorized form, decomposing them into base geometry and time-dependent relative motion. Extensive experiments demonstrate that 4RC outperforms prior and concurrent methods across a wide range of 4D reconstruction tasks. *Project Page:* https://yihangluo.com/projects/4RC/.

[1]S-Lab, Nanyang Technological University [2]VGG, University of Oxford [3]ACE Robotics.

*Proceedings of the 43rd International Conference on Machine Learning*, Seoul, South Korea. PMLR 306, 2026. Copyright 2026 by the author(s).

## 1. Introduction

3D reconstruction has seen remarkable progress over the past decades. Classical geometric pipelines such as Structure-from-Motion (SfM) (Schönberger & Frahm, 2016) and Multi-View Stereo (MVS) (Yao et al., 2018; 2019; Schönberger et al., 2016) established a solid foundation. More recently, learning-based approaches, exemplified by DUSt3R-like pointmap predictor (Wang et al., 2024b; Leroy et al., 2024; Wang et al., 2025b;a;d; Lin et al., 2026; Lan et al., 2026) have enabled direct feed-forward inference of dense 3D geometry, advancing general-purpose 3D perception in terms of efficiency, scalability, and generalization.

Despite this progress, existing approaches largely focus on static geometry, while real-world scenes are inherently dynamic. A truly general visual perception system must therefore reason not only about 3D structure, but also about how the scene evolves over time. This motivates the task of *4D reconstruction*, which aims to jointly model 3D geometry and motion. Such a representation is fundamental for applications ranging from video synthesis (Gu et al., 2025; Wu et al., 2024; Lee et al., 2025b) and scene understanding to robotics (Lee et al., 2025a; Huang et al., 2026), where reasoning about object trajectories, deformations, and interactions is essential.

Existing approaches to 4D reconstruction, however, remain fragmented and limited in flexibility. A common strategy decomposes the problem into sequential subtasks, typi-

cally separating motion estimation from 3D reconstruction. For example, SpatialTracker (Xiao et al., 2024; 2025) performs reconstruction and tracking in a staged manner, relying on iterative refinement, and producing only sparse 3D trajectories. MonST3R (Zhang et al., 2025b) further requires post-hoc optimization to establish correspondences across time. Although recent feed-forward methods such as ST4RTrack (Feng et al., 2025) and Dynamic Point Map (Sucar et al., 2025) pioneer direct 4D prediction, they are restricted to pairwise views and thus struggle to model long-term and complex motion. Concurrently, TraceAnything (Liu et al., 2026a) represents motion using Bézier curves, enabling long-range 3D trajectory tracking, but often at a cost of reduced geometry quality. Any4D (Karhade et al., 2025) supports feed-forward 3D reconstruction, but only predicts scene flow for the first frame and is unable to model 3D motion for the remaining frames. V-DPM (Sucar et al., 2026) extends VGGT to 4D, but suffers from slow inference and limited flexibility at inference.

Motivated by these limitations, we investigate whether a unified, feed-forward model can enable complete and flexible 4D prediction. In this work, we propose 4RC, a unified feed-forward approach for 4D reconstruction from monocular videos. Unlike previous approaches that require multiple stages, 4RC learns a holistic and compact 4D representation that jointly encodes scene geometry and motion across the entire video sequence. This representation serves as a centralized 4D latent from which geometry and motion can be efficiently queried and decoded. Instead of directly reconstructing a full 3D point cloud for each frame at each timestamp, we adopt a compact factorized output formulation. Specifically, we represent each frame with a viewpoint-invariant *base geometry* together with time-dependent *relative motion*, parameterized as 3D displacements. By querying the model at different timestamps, 4RC can recover both geometry and motion information, such as point trajectories between any frame and any target time. This design enables both flexible and efficient 4D reconstruction.

Our contributions can be summarized as follows:

- A unified feed-forward transformer framework for 4D reconstruction from monocular videos, which jointly models 3D geometry and motion within a single network, eliminating the need for auxiliary estimators or per-scene optimization.

- An *encode-once, query-anywhere and anytime* paradigm built upon a compact 4D latent representation. This allows our conditional decoder to flexibly retrieve dense 3D geometry and motion for arbitrary query frames at any target timestamp.

- A minimally factorized 4D representation that decomposes each frame into a viewpoint-invariant base ge-

ometry and time-dependent relative motion, enabling unified and flexible reconstruction of dynamic scenes.

Extensive experiments demonstrate that 4RC achieves competitive performance on standard benchmarks across a wide range of 3D and 4D reconstruction tasks, including camera pose estimation, video depth prediction, point cloud reconstruction, 3D point tracking, and dense motion modeling.

## 2. Related Work

**Feed-forward 3D Reconstruction.** Reconstructing 3D geometry from 2D images is a long-standing problem in computer vision. Traditional pipelines such as SfM (Schönberger & Frahm, 2016) and MVS (Schönberger et al., 2016; Yao et al., 2018; 2019) recover camera parameters and dense geometry through multi-stage optimization, achieving strong performance but at high computational cost. Recent work has shifted toward feed-forward 3D reconstruction, aiming to replace these complex pipelines with a single neural network that directly predicts 3D attributes. DUSt3R (Wang et al., 2024b) demonstrates that dense stereo reconstruction can be achieved in one forward pass, while VGGT (Wang et al., 2025a) further unifies camera pose estimation and depth prediction across multiple views using a transformer backbone. These methods highlight that, given sufficient data and model capacity, feed-forward architectures can effectively solve static 3D reconstruction. Extensions to dynamic settings, such as MonST3R (Zhang et al., 2025b), Pi3 (Wang et al., 2025d), DA3 (Lin et al., 2026) and related approaches (Wang et al., 2025b; Lan et al., 2026), jointly estimate camera parameters and per-frame geometry from dynamic data. Despite operating on dynamic scenes, these methods only reconstruct geometry for each view and thus require separate pipelines to explicitly model 3D motion or temporal correspondence.

**Point Tracking.** Modeling motion over time has traditionally been studied through optical flow (Sun et al., 2010) and point tracking (Harley et al., 2022). Optical flow methods (Sun et al., 2018; Hui et al., 2018; Teed & Deng, 2020) estimate dense pixel-wise displacements between adjacent frames. These methods are typically limited to short temporal windows and often suffer from drift errors when applied to long video sequences (Zhou et al., 2023). To address long-range correspondence, 2D point tracking methods aim to track sparse points across entire videos. PIPs (Harley et al., 2022) introduced a deep tracking framework for point tracking, followed by TAP-Net (Doersch et al., 2022), TAPIR (Doersch et al., 2023), and CoTracker (Karaev et al., 2023a), which rely on correlation-based matching and iterative updates to propagate tracks over time. These approaches operate purely in 2D and typically depend on carefully designed matching and update mechanisms. Recent 3D point tracking approaches extend this paradigm by decoupling

geometry reconstruction from motion modeling. Spatial-Tracker (Xiao et al., 2024), and subsequent methods (Ngo et al., 2024; Xiao et al., 2025; Zhang et al., 2025a) combine a pre-trained depth estimator with a lifted 2D tracking pipeline (Karaev et al., 2023a) to operate in 3D. Despite enabling 3D tracking, their multi-stage pipelines remain limited in efficiency and flexibility, and they do not learn a unified spatiotemporal representation. In contrast, 4RC directly models dense geometry and motion jointly within a unified feed-forward framework, without decoupled stages or tracking heuristics.

**4D Reconstruction.** The goal of 4D reconstruction is to recover a representation that captures both the 3D structure of a scene and how it evolves over time. Early methods (Wang et al., 2023a; 2024a; Lei et al., 2024; Wang et al., 2025c) typically formulate this problem as test-time optimization, which can produce high-quality results but requires costly per-scene optimization. Recent efforts have gradually shifted toward feed-forward formulations of 4D reconstruction. St4RTrack (Feng et al., 2025) predicts point maps for pairs of views, jointly encoding static geometry and dynamic motion; however, its pairwise formulation inherently limits the temporal range of the reconstruction. We also acknowledge several recent concurrent works that explore feed-forward formulations for 4D reconstruction. TraceAnything (Liu et al., 2026a) represents scenes using continuous trajectory fields parameterized by Bézier curves. Although this formulation enables smooth and long-range motion modeling, it often struggles to represent complex or high-frequency dynamics and may compromise geometric accuracy. Any4D (Karhade et al., 2025) jointly predicts scene flow and 3D geometry from a canonical reference view, but lacks the flexibility to infer motion originating from arbitrary viewpoints. Similarly, V-DPM (Sucar et al., 2026) extends VGGT to dynamic settings, but relies on an inflexible decoding scheme that aggregates information from all views, leading to high computational costs. Concurrently, D4RT (Zhang et al., 2026) introduces a unified model for 2D and 3D point tracking. Specifically, D4RT first encodes the entire video into a global scene representation using a self-attention encoder, and then answers spatio-temporal per-point queries through a lightweight cross-attention decoder. Likewise, our method, 4RC, employs a flexible query-based decoder that efficiently recovers complete and dense 4D attributes for any view at any timestamp, without expensive per-point computation.

## 3. Method

Our goal is to develop a unified and feed-forward model, 4RC, that takes a monocular video as input and reconstructs the full underlying 4D attributes of the scene. The core of our approach lies in encoding the entire video sequence into

a compact 4D representation, which can then be queried on-demand to decode the geometry and motion of any query frame at any target timestamp, as illustrated in Figure 2.

### 3.1. Problem Formulation

Given a monocular video sequence $\mathcal{V} = \{I_i\}_{i=1}^N$, where $I_i \in \mathbb{R}^{H \times W \times 3}$ denotes the RGB frame captured at timestamp $t_i$ and $N$ is the total number of frames, our goal is to recover the full 4D attributes of the scene, capturing both its 3D structure and temporal evolution. Specifically, for any query frame $I_i$ and an arbitrary target timestamp $\tau \in \{t_i\}_{i=1}^N$, we define a time-indexed 3D point map:

$$P_i^{t_i \to \tau} \in \mathbb{R}^{H \times W \times 3}, \tag{1}$$

which represents the 3D positions of points observed in frame $I_i$ as they appear at time $\tau$. When $\tau = t_i$, $P_i^{t_i \to \tau}$ corresponds to the static 3D geometry of the frame. When $\tau \neq t_i$, it describes the dynamic time-dependent point maps of the scene by mapping the points from the source frame to their locations at the target time.

**Factorized 4D Attributes.** Directly predicting point maps $P_i^{t_i \to \tau}$ for all possible $(i, \tau)$ pairs is redundant and intractable. Once the underlying 3D geometry at the source time is known, the geometry at other times can be expressed through relative motion. We therefore adopt a factorized representation:

$$P_i^{t_i \to \tau} = P_i^{t_i} + \Delta P_i^{t_i \to \tau}, \tag{2}$$

where $P_i^{t_i}$ denotes the base 3D geometry at time $t_i$, and $\Delta P_i^{t_i \to \tau}$ represents the 3D displacement from time $t_i$ to $\tau$.

This formulation offers both *conceptual* and *practical* advantages. The base geometry $P_i^{t_i}$ is reconstructed from image $I_i$ under the perspective camera model, a property that allows us to directly leverage recent advances of effective geometry representation in monocular 3D reconstruction (Lin et al., 2026). Meanwhile, the displacement field $\Delta P_i^{t_i \to \tau}$ explicitly captures temporal motion. This provides clear motion cues that are useful for downstream applications, while avoiding the need to re-predict complex geometry at every time step. As a result, the representation remains temporally consistent, especially in static regions and under rigid motion. Unless otherwise stated, all point maps are viewpoint-invariant and expressed in a world coordinate system defined by the camera of the first frame (Wang et al., 2024b; 2025b;a; Lin et al., 2026).

**Relation with Other Work.** The key distinction between 4RC and several prior or concurrent approaches lies in the flexibility and completeness of our 4D output. Recent feed-forward 3D reconstruction methods focus solely on predicting the base 3D geometry for each input frame, i.e., $P_i^{t_i}$, and thus fail to capture the motion within the scene.

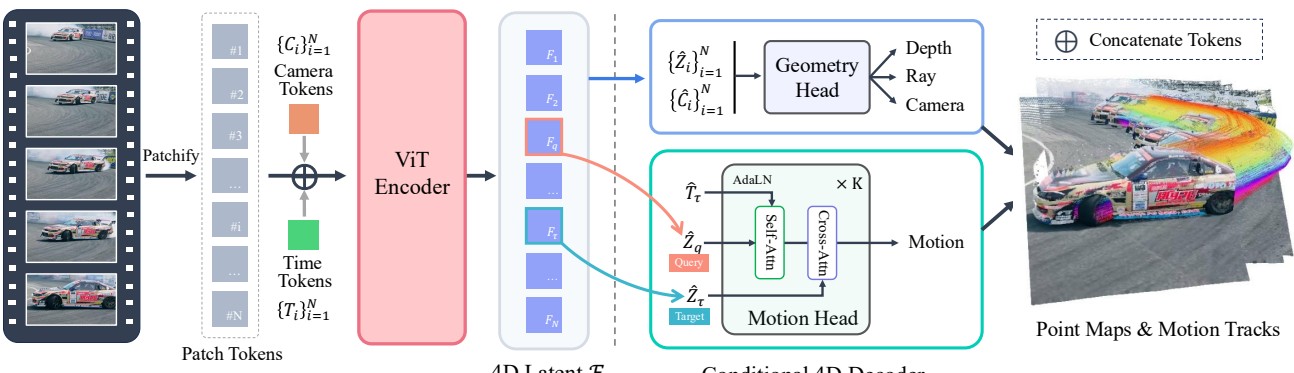

*Figure 2.* **Overall architecture of 4RC.** Video frames are patchified and augmented with camera and time tokens, then jointly encoded by a single transformer into a compact 4D latent representation $\mathcal{F}$, from which a conditional decoder with disentangled geometry and motion heads enables flexible querying of 3D geometry and motion for arbitrary source views at arbitrary target timestamps.

Traditional 3D point tracking methods, on the other hand, estimate sparse trajectories initialized from selected points and therefore cannot recover dense 4D geometry. Concurrent feed-forward 4D reconstruction methods also exhibit limitations in motion modeling. St4RTrack is restricted to pairwise motion. TraceAnything models trajectory fields using Bézier curves, which limits its ability to capture accurate geometry and complex motion. Any4D predicts motion only relative to the first frame, i.e., $P_1^{t_1 \to \tau}$ with $\tau \in \{t_i\}_{i=1}^N$, and therefore cannot support motion queries from other source frames. V-DPM regresses the point map $P_i^{t_i \to \tau}$ for all source frames $i \in \{1, \dots, N\}$ at a given target timestamp $\tau$, by attending to all frames jointly, which incurs substantial computational overhead and limits inference flexibility. In contrast, 4RC enables flexibly querying dense 3D motion from any single source frame to any target timestamp within a unified and fully feed-forward framework.

### 3.2. 4D Representation Encoder

The encoder $\mathcal{E}$ processes the input video $\mathcal{V}$ to produce a unified 4D representation:

$$\mathcal{F} = \mathcal{E}(\mathcal{V}). \tag{3}$$

We adopt a plain ViT-based transformer architecture that alternates between frame-wise self-attention and global self-attention. Similar to the camera token in VGGT (Wang et al., 2025a), which primarily encodes camera geometry information for subsequent decoding, we further append each view's patchified tokens with a dedicated time token $T_i$. This time token aggregates temporal information for that view and serves as a conditioning signal for target-time motion decoding, as described in Section 3.3. The encoder produces a unified spatio-temporal latent representation $\mathcal{F} = \{F_i\}_{i=1}^N$. Each $F_i = \{\hat{Z}_{i,j}\}_{j=1}^M \cup \{\hat{C}_i\} \cup \{\hat{T}_i\}$ consists of $M$ patch tokens $\hat{Z}_{i,j} \in \mathbb{R}^D$ corresponding to the $i$-th frame, together with an encoded camera token $\hat{C}_i$ and a time token $\hat{T}_i$. We

treat $\mathcal{F}$ as an ordered sequence of frame-level token sets.

### 3.3. Conditional 4D Decoder

**Geometry Head.** To recover the base geometry for each input frame, we use a geometry decoder $\mathcal{D}_\mathrm{g}$. Given the encoded spatial tokens $\hat{Z}_i$ and camera tokens $\hat{C}_i$, the geometry decoder predicts per-frame depth and rays, together with camera parameters:

$$\left( \hat{D}_i, \hat{R}_i, \hat{\theta}_i \right) = \mathcal{D}_\mathrm{g}\left( \hat{Z}_i, \hat{C}_i \right), \tag{4}$$

where $\hat{D}_i \in \mathbb{R}^{H \times W}$ is the depth map, $\hat{R}_i \in \mathbb{R}^{\frac{1}{2}H \times \frac{1}{2}W \times 6}$ is the ray map, and $\hat{\theta}_i$ denotes the camera parameters (i.e., field of view, rotation, and translation). The base point map $P_i^{t_i}$ is then obtained from $(\hat{D}_i, \hat{R}_i, \theta_i)$ under the perspective camera model. The geometry decoder $\mathcal{D}_\mathrm{g}$ follows a dual-DPT (Ranftl et al., 2021; Lin et al., 2026) design with a lightweight camera head.

**Motion Head.** To recover motion for any query frame $I_q$ at a target timestamp $\tau$, we use a lightweight transformer-based motion decoder $\mathcal{D}_\mathrm{m}$ with $K$ layers of alternating self-attention and cross-attention. We initialize the query tokens $\hat{Z}_q$ from the encoder output $\mathcal{F}$. The decoder outputs a dense 3D displacement field:

$$\Delta \hat{P}_q^{t_q \to \tau} = \mathcal{D}_\mathrm{m}\left( \hat{Z}_q, \hat{T}_\tau, \hat{Z}_\tau \right). \tag{5}$$

Specifically, to condition on the target time, we inject time embedding $\hat{T}_\tau$ via Adaptive Layer Normalization (AdaLN) (Perez et al., 2018) in the self-attention blocks, and then apply cross-attention to the target spatial token set $\hat{Z}_\tau$. This design supports dense motion estimation and point tracking while remaining compatible with our per-frame geometry decoding.

### 3.4. Training Scheme

We train 4RC in an end-to-end manner with joint supervision over geometry and motion attributes. Following prior works (Wang et al., 2025a; Lin et al., 2026), we normalize the ground-truth scene scale such that the average Euclidean distance of all valid 3D points to the origin is 1. The overall training objective is defined as:

$$\mathcal{L} = \mathcal{L}_{\text{depth}} + \mathcal{L}_{\text{ray}} + \mathcal{L}_{\text{cam}} + \mathcal{L}_{\text{motion}}. \tag{6}$$

For all loss terms except the camera parameter loss $\mathcal{L}_{\text{cam}}$, we adopt an aleatoric uncertainty formulation (Wang et al., 2024b). We denote the loss function as $\ell(\hat{\mathbf{y}}, \mathbf{y}, \boldsymbol{\Sigma})$, where $\boldsymbol{\Sigma}$ represents the predicted pixel-wise uncertainty map, which adaptively down-weights unreliable regions during training.

To better supervise both geometry and motion, we apply gradient-based constraints (Lin et al., 2026) in the spatial and temporal domains separately. For geometry learning, we enforce spatial smoothness on the predicted depth maps $\hat{\mathbf{D}} = \{\hat{D}_i\}$ by applying image-space gradients $\nabla_{\mathbf{x}}$. The depth loss is formulated as:

$$\mathcal{L}_{\text{depth}} = \ell(\hat{\mathbf{D}}, \mathbf{D}, \boldsymbol{\Sigma}_D) + \ell(\nabla_{\mathbf{x}}\hat{\mathbf{D}}, \nabla_{\mathbf{x}}\mathbf{D}, \boldsymbol{\Sigma}_D). \tag{7}$$

Similarly, the motion loss supervises the displacement field $\Delta\mathbf{P}$, but we incorporate an additional temporal gradient term $\nabla_t$ that constrains the first-order temporal derivative of the displacement (i.e., velocity) to encourage temporally consistent motion behavior:

$$\begin{aligned} \mathcal{L}_{\text{motion}} = \ &\ell(\Delta\hat{\mathbf{P}}, \Delta\mathbf{P}, \boldsymbol{\Sigma}_M) \\ &+ \ell(\nabla_t\Delta\hat{\mathbf{P}}, \nabla_t\Delta\mathbf{P}, \boldsymbol{\Sigma}_M). \end{aligned} \tag{8}$$

## 4. Experiments

We conduct extensive experiments to evaluate the effectiveness of 4RC on standard 4D reconstruction tasks. We compare against established state-of-the-art methods as well as concurrent work for completeness, and further perform ablation studies to analyze the contribution of key design components in our framework.

### 4.1. Training Setup

**Datasets.** We train 4RC on a diverse collection of large-scale public datasets, covering both dynamic and static scenes, as well as synthetic and real-world videos. Specifically, our training data includes PointOdyssey (Zheng et al., 2023), Dynamic Replica (Karaev et al., 2023b), Kubric (Greff et al., 2022), Waymo (Sun et al., 2020), DL3DV (Ling et al., 2024), ScanNet++ (Yeshwanth et al., 2023), and MVS-Synth (Huang et al., 2018). These datasets jointly provide rich supervision for geometry, motion, and camera poses

under varied scene layouts and motion patterns. Detailed dataset statistics are provided in the appendix.

**Implementation Details.** Our encoder adopts a single Vision Transformer based on DINOv2 (Oquab et al., 2023). The motion decoder is lightweight, consisting of $K = 4$ layers of self-attention and cross-attention. We initialize both the encoder and the geometry decoder with pretrained weights from DA3 (Lin et al., 2026), which is trained on large-scale 3D data and provides strong geometric priors. During training, input images are resized to a randomly sampled resolution, with the longer side up to 504 pixels. The aspect ratio is uniformly sampled from $[0.5, 2.0]$ to improve generalizability. The training sequence length $N$ is randomly sampled from $[2, 18]$ views, with longer sequences facilitating larger and more complex motions. To avoid the quadratic cost of computing all $N^2$ motion pairs, we randomly sample one query view per iteration and predict its motion in $N$ different timesteps during training. Standard data augmentations including color jittering and Gaussian blur are applied. The model is trained end-to-end using the training loss described in Section 3.4. We use the AdamW optimizer (Kingma & Ba, 2015; Loshchilov & Hutter, 2019) for 50 epochs with a cosine learning rate schedule. Training is performed on 16 A100 GPUs with a batch size of 1 per GPU. Additional implementation details and hyperparameters are provided in the appendix.

### 4.2. 4D Reconstruction

**Qualitative Results.** Figure 3 provides qualitative comparisons of 4RC in modeling 3D tracking. These visual results demonstrate the effectiveness of our method in handling complex motion patterns, such as occlusions, non-rigid motion, and large movements. We further evaluate our method on diverse in-the-wild videos in Figure 4, demonstrating its strong performance on both static and dynamic scenes.

**Dense Tracking.** To demonstrate the capability of our method to track dense motion from arbitrary query views, we first quantitatively evaluate dense 3D tracking by sampling 24 frames from the Kubric and Waymo test sets, with 50 samples each, using the middle view (i.e., the 11th frame) as the query. Traditional point tracking methods fail on dense tracking due to out-of-memory issues, while Any4D can only predict the motion field for the first view. We report both the *Average Percentage of Points* (APD) within a threshold and the *End-Point Error* (EPE) after global Sim(3) alignment with RANSAC. As shown in Table 1(a), 4RC achieves state-of-the-art performance among concurrent 4D reconstruction methods on both datasets. On the challenging Waymo dataset, which contains highly dynamic scenes, our method substantially outperforms the concurrent method V-DPM, resulting in a 36% gain in APD. Notably, our method uses flexible per-frame decoding, in contrast to V-DPM's computationally expensive global aggregation decoding.

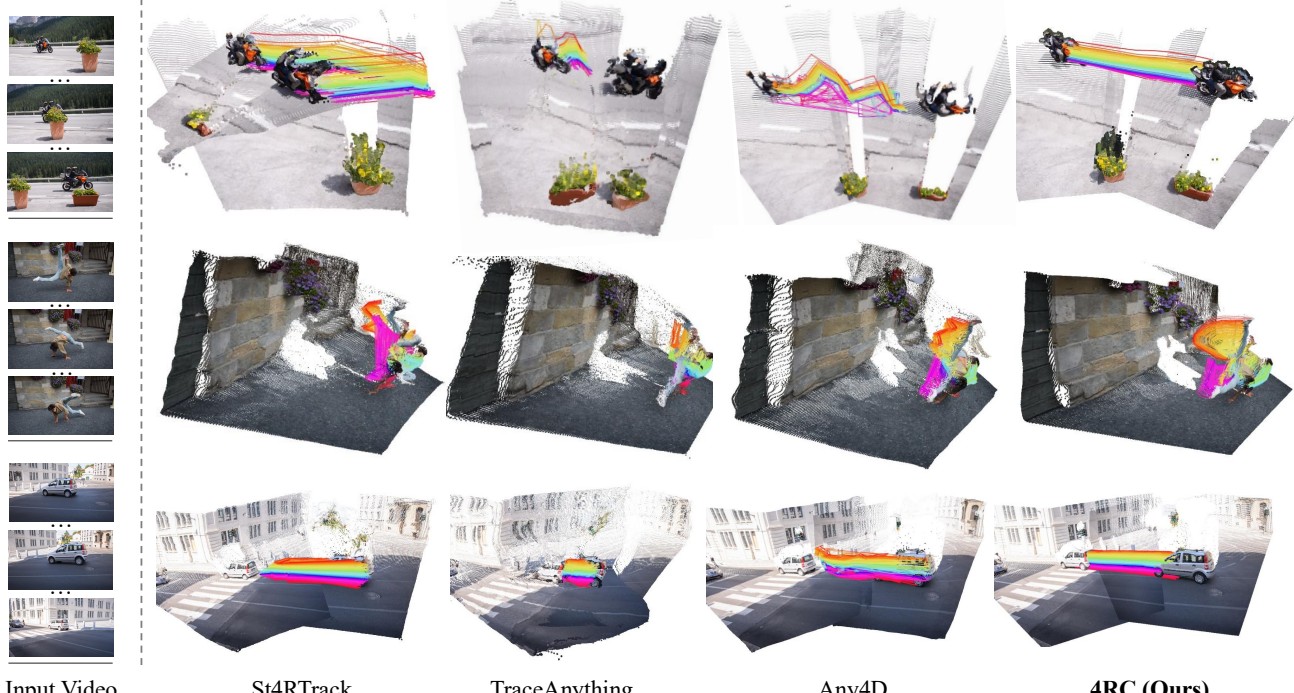

Input Video     St4RTrack     TraceAnything     Any4D     **4RC (Ours)**

*Figure 3.* **Qualitative comparison of dynamic tracking on DAVIS** (Perazzi et al., 2016). We visualize the dynamic reconstruction results, including the geometry at the first and last frames, as well as the dynamic object trajectories rendered as rainbow-colored paths from the first view. As shown in the top example, our method successfully handles occlusion when the motorcycle becomes temporarily invisible. In contrast, the two-view method St4RTrack lacks global temporal context and therefore predicts an incorrect trajectory. In the second and third examples, our method accurately reconstructs complex and large-scale motions while preserving high-quality geometry, while other methods produce inconsistent motion trajectories and degraded geometry.

*Table 1.* **4D reconstruction evaluation on tracking**. We evaluate our method on dense-view tracking (a), as well as sparse-view tracking (b) on dynamic datasets. Our method demonstrates state-of-the-art capability in dense tracking from arbitrary views compared to concurrent 4D reconstruction methods, and also achieves strong performance on the sparse point tracking setting, even when compared to tracking-specific methods. The top-2 results are highlighted as best and second .

| Method | (a) Dense Tracking | | | | (b) Sparse Point Tracking | | | | | | | |
| | Kubric | | Waymo | | PO | | DR | | ADT | | PStudio | |
| | APD ↑ | EPE ↓ | APD ↑ | EPE ↓ | APD ↑ | EPE ↓ | APD ↑ | EPE ↓ | APD ↑ | EPE ↓ | APD ↑ | EPE ↓ |
|---|---|---|---|---|---|---|---|---|---|---|---|---|
| VGGT + CoTracker3 (Karaev et al., 2024) | - | - | - | - | 63.19 | 0.5890 | 80.93 | 0.2417 | 77.81 | 0.3015 | 78.11 | 0.2715 |
| SpatialTrackerV2 (Xiao et al., 2025) | - | - | - | - | 73.66 | 0.3944 | 80.87 | 0.2218 | 95.48 | 0.0594 | 85.63 | 0.1583 |
| St4RTrack (Feng et al., 2025) | 50.65 | 3.938 | 19.98 | 6.359 | 71.64 | 0.3101 | 78.36 | 0.2367 | 82.79 | 0.2279 | 74.05 | 0.2537 |
| TraceAnything (Liu et al., 2026a) | 59.98 | 1.808 | 21.25 | 4.313 | 52.02 | 0.9154 | 68.28 | 0.5060 | 82.77 | 0.1998 | 74.15 | 0.2926 |
| Any4D (Karhade et al., 2025) | - | - | - | - | 71.47 | 0.3642 | 81.28 | 0.2171 | 73.83 | 0.3114 | 78.76 | 0.2088 |
| V-DPM (Sucar et al., 2026) | 71.12 | 2.849 | 41.44 | 1.948 | 83.36 | 0.1955 | 83.04 | 0.1901 | 80.80 | 0.2357 | 89.59 | 0.1165 |
| **4RC (Ours)** | 85.44 | 1.022 | 56.63 | 1.611 | 85.86 | 0.2498 | 88.65 | 0.1484 | 87.82 | 0.1480 | 87.32 | 0.1304 |

**Sparse Point Tracking.** We then evaluate 4RC on 3D sparse point tracking, which measures sparse motion relative to the first frame, although our method can fully capture dense motion. Following the WorldTrack benchmark (Feng et al., 2025), tracking performance is assessed in the world coordinate system. The benchmark includes two datasets, Aerial Digital Twin (ADT) (Pan et al., 2023) and Panoptic Studio (PStudio) (Joo et al., 2019) from TAPVid-3D (Koppula et al., 2024), as well as two test sets from PointOdyssey

(PO) and Dynamic Replica (DR). We compare our method against tracking-specific methods CoTracker3 (Karaev et al., 2024) and SpatialTrackerV2 (Xiao et al., 2025), along with concurrent 4D reconstruction methods. The predicted trajectory is aligned to the ground truth using a global Sim(3) transformation via RANSAC. As shown in Table 1(b), 4RC achieves strong performance even when compared with methods specifically designed for point tracking, outperforming SpatialTrackerV2 on 3 out of 4 datasets.

*Table 2.* **Camera pose estimation and multi-view 3D reconstruction evaluation.** We compare our method with both 3D reconstruction approaches and concurrent 4D reconstruction methods. Our approach achieves state-of-the-art performance among 4D methods, while remaining competitive with 3D reconstruction methods Pi3, without exclusive training on large-scale reconstruction datasets.

| Method | (a) Camera Pose Estimation | | | | | | (b) Multi-View 3D Reconstruction | | | | | |
| | TUM-dynamics | | | ScanNet | | | 7-Scenes | | | NRGBD | | |
| | ATE ↓ | RPE$_t$ ↓ | RPE$_r$ ↓ | ATE ↓ | RPE$_t$ ↓ | RPE$_r$ ↓ | Acc ↓ | Comp ↓ | NC ↑ | Acc ↓ | Comp ↓ | NC ↑ |
|---|---|---|---|---|---|---|---|---|---|---|---|---|
| DUSt3R (Wang et al., 2024b) | 0.083 | 0.017 | 3.567 | 0.081 | 0.028 | 0.784 | 0.146 | 0.181 | 0.736 | 0.144 | 0.154 | 0.870 |
| MASt3R (Leroy et al., 2024) | 0.038 | 0.012 | 0.448 | 0.078 | 0.020 | 0.475 | 0.185 | 0.180 | 0.701 | 0.085 | 0.063 | 0.794 |
| MonST3R (Zhang et al., 2025b) | 0.098 | 0.019 | 0.935 | 0.077 | 0.018 | 0.529 | 0.248 | 0.266 | 0.672 | 0.272 | 0.287 | 0.758 |
| Spann3R (Wang & Agapito, 2024) | 0.056 | 0.021 | 0.591 | 0.096 | 0.023 | 0.661 | 0.298 | 0.205 | 0.650 | 0.416 | 0.417 | 0.684 |
| CUT3R (Wang et al., 2025b) | 0.046 | 0.015 | 0.473 | 0.099 | 0.022 | 0.600 | 0.126 | 0.154 | 0.727 | 0.099 | 0.076 | 0.837 |
| VGGT (Wang et al., 2025a) | 0.012 | 0.010 | 0.311 | 0.036 | 0.015 | 0.376 | 0.087 | 0.091 | 0.787 | 0.073 | 0.077 | 0.910 |
| Pi3 (Wang et al., 2025d) | 0.014 | 0.009 | 0.309 | 0.031 | 0.013 | 0.346 | 0.044 | 0.063 | 0.758 | 0.022 | 0.025 | 0.911 |
| St4RTrack (Feng et al., 2025) | - | - | - | - | - | - | 0.240 | 0.234 | 0.681 | 0.241 | 0.219 | 0.754 |
| TraceAnything (Liu et al., 2026a) | - | - | - | - | - | - | 0.232 | 0.359 | 0.584 | 0.347 | 0.527 | 0.643 |
| Any4D (Karhade et al., 2025) | 0.030 | 0.023 | 0.463 | 0.074 | 0.035 | 1.076 | 0.141 | 0.177 | 0.738 | 0.081 | 0.072 | 0.847 |
| V-DPM (Sucar et al., 2026) | 0.014 | 0.010 | 0.318 | 0.035 | 0.014 | 0.410 | 0.097 | 0.124 | 0.772 | 0.056 | 0.060 | 0.897 |
| **4RC (Ours)** | 0.010 | 0.008 | 0.314 | 0.032 | 0.012 | 0.437 | 0.034 | 0.051 | 0.783 | 0.036 | 0.034 | 0.912 |

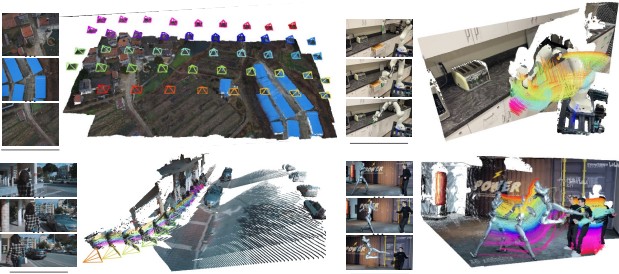

*Figure 4.* **Visualization of in-the-wild examples**. 4RC demonstrates accurate geometry reconstruction and motion modeling in both static and dynamic scenes.

### 4.3. 3D Reconstruction

**Camera Pose Estimation.** We evaluate camera pose estimation on the Sintel (Butler et al., 2012), TUM-dynamics (Sturm et al., 2012), and ScanNet (Dai et al., 2017) datasets. Performance is measured using *Absolute Translation Error* (ATE), *Relative Translation Error* (RPE$_t$), and *Relative Rotation Error* (RPE$_r$), all computed after global Sim(3) alignment with the ground truth, following established protocols (Teed & Deng, 2021; Zhang et al., 2025b; Wang et al., 2025b). Table 2 (a) shows that 4RC achieves top-tier camera pose estimation and reconstruction quality within a single unified model. On the challenging TUM-dynamics dataset, 4RC attains the best ATE and RPE$_t$ among all methods, including specialized 3D reconstruction methods such as Pi3, which are trained on much larger datasets. This demonstrates that our unified 4D representation is effective for both motion modeling and producing accurate camera trajectories. Notably, 4RC achieves the best performance among concurrent feed-forward 4D reconstruction methods. We exclude St4RTrack and TraceAnything as

*Table 3.* **Video depth estimation on the Bonn and Sintel datasets.** We compare methods that explicitly predict video depth.

| Method | Bonn | | Sintel | |
| | Rel ↓ | δ < 1.25 ↑ | Rel ↓ | δ < 1.25 ↑ |
|---|---|---|---|---|
| DUSt3R (Wang et al., 2024b) | 0.155 | 83.3 | 0.656 | 45.2 |
| MASt3R (Leroy et al., 2024) | 0.252 | 70.1 | 0.641 | 43.9 |
| MonST3R (Zhang et al., 2025b) | 0.067 | 96.3 | 0.378 | 55.8 |
| Spann3R (Wang & Agapito, 2024) | 0.144 | 81.3 | 0.622 | 42.6 |
| CUT3R (Wang et al., 2025b) | 0.078 | 93.7 | 0.421 | 47.9 |
| Fast3R (Yang et al., 2025) | 0.193 | 77.5 | 0.653 | 44.9 |
| VGGT (Wang et al., 2025a) | 0.055 | 97.1 | 0.297 | 68.8 |
| Pi3 (Wang et al., 2025d) | 0.050 | 97.4 | 0.246 | 67.7 |
| **4RC (Ours)** | 0.051 | 97.4 | 0.311 | 62.2 |

they do not explicitly estimate camera poses.

**Multi-View Reconstruction.** Following prior work (Wang & Agapito, 2024; Wang et al., 2025b; 2024b), we evaluate scene-level multi-view 3D reconstruction on the 7-Scenes (Shotton et al., 2013) and NRGBD (Azinović et al., 2022) datasets. Reconstruction quality is measured using *Accuracy* (Acc), *Completeness* (Comp), and *Normal Consistency* (NC). Quantitative results are reported in Table 2 (b). 4RC achieves the best performance among 4D reconstruction methods, attaining the highest Acc/Comp on 7-Scenes and the best NC on NRGBD. This highlights the effectiveness of our proposed design. For example, we obtain 0.034 accuracy on 7-Scenes, far better than TraceAnything's 0.240; the latter jointly models geometry and motion in a trajectory field, which often compromises geometric quality.

**Depth Estimation.** We also evaluate video depth estimation on Sintel (Butler et al., 2012) and Bonn (Palazzolo et al., 2019) datasets. Following prior work (Wang et al.,

*Table 4.* **Ablation of our motion head design and factorized motion.** In (a), we evaluate the effectiveness of our motion head design by removing each component. (b) shows that representing the motion output in a factorized form performs better than directly predicting the point cloud.

| Methods | Kubric | | Waymo | |
|---|---|---|---|---|
| | APD ↑ | EPE ↓ | APD ↑ | EPE ↓ |
| **4RC (Ours)** | **85.44** | **1.022** | **56.63** | **1.611** |
| *(a) Motion Head Design* | | | | |
| (i)  w/o Cross Attn. | 80.83 | 1.136 | 54.19 | 1.618 |
| (ii)  w/o Self Attn. | 80.57 | 1.127 | 53.50 | 1.686 |
| (iii) w/o AdaLN | 82.51 | 1.105 | 56.11 | 1.689 |
| *(b) Factorized Motion* | | | | |
| (i)  Points (World) | 74.64 | 1.412 | 37.08 | 2.359 |
| (ii) Points (Local) | 70.70 | 1.547 | 19.55 | 3.226 |

2025b), predicted depth maps are aligned to the ground truth using a per-sequence scale factor. While most existing 4D reconstruction methods do not explicitly output depth and therefore cannot be directly evaluated on depth benchmarks, 4RC includes an explicit depth prediction as part of its factorized 4D representation. On the Bonn dataset, 4RC achieves the best $\delta < 1.25$ score and matches the second-best Rel. On Sintel, there is a small gap compared to specialized 3D reconstruction methods such as Pi3, which are trained exclusively on large-scale 3D datasets that are more than twice the size of our training datasets.

## 4.4. Ablation Studies

We conduct ablation studies to evaluate the key design choices in 4RC, focusing on the motion head and the factorized motion representation.

**Motion Head Design.** Our motion head enables motion querying from arbitrary input views at arbitrary target timestamps. To analyze the contribution of each component in the motion head, we construct several variants by removing individual modules: (i) cross-attention between query tokens and target-time latent features, (ii) self-attention, and (iii) time-token conditioned AdaLN. All variants use the same number of layers and have comparable parameter sizes. As shown in Table 4 (a), removing any component consistently degrades performance, indicating that all modules are necessary for effective motion decoding. Among them, removing either attention module results in the largest performance drop. In Figure 5, we also quantitatively observe that without cross-attention, the decoder struggles to model complex non-rigid motions, such as hand and leg movements, producing over-smoothed trajectories that do not align with the true motion. This suggests that self-attention and adaptive normalization alone are insufficient for handling large and detailed temporal displacements, and direct access to target-time features is critical for accurate motion estimation.

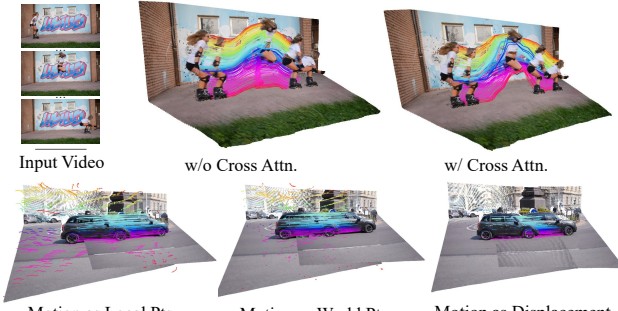

Input Video    w/o Cross Attn.    w/ Cross Attn.

Motion as Local Pts.    Motion as World Pts.    Motion as Displacement

*Figure 5.* **Qualitative ablation visualizations.** The first row shows the effectiveness of cross-attention in the motion head: without it, although the model outputs rough trajectories, it fails to capture fine details such as the motion of the girl's legs and hands when she is at the peak of a jump. The second row illustrates that outputting motion as point clouds can lead to inconsistent trajectories as it requires re-predicting base geometry for each time step.

**Factorized Motion.** We further evaluate the effectiveness of our factorized motion representation by comparing it with alternative output parameterizations commonly used in 3D reconstruction (Wang et al., 2025a;d). Specifically, we replace our displacement-based formulation with two point-based variants: directly predicting 3D coordinates in (i) a shared world coordinate system, or (ii) each view's own camera coordinate system. As reported in Table 4 (b), both point-based variants perform worse than our factorized representation. This performance gap arises mainly from differences in representation. Direct point prediction entangles geometry and motion in a single output space, forcing the network to jointly learn shape and temporal correspondences, which significantly increases learning difficulty. Qualitative results in Figure 5 further support this observation. Our formulation explicitly decouples static geometry from time-dependent motion via displacement fields, reducing unnecessary recomputation of geometry and improving temporal consistency.

## 5. Conclusion

We present 4RC, a unified feed-forward transformer framework for 4D reconstruction from monocular videos. Central to our approach is a novel *encode-once, query-anywhere and anytime* paradigm, in which a compact 4D representation of the entire video is learned once and subsequently queried to recover geometry and motion at arbitrary time instances. This paradigm effectively bridges the global spatio-temporal modeling with flexible, on-demand query-based reconstruction, achieving both accurate 4D reconstruction and high efficiency. Extensive experiments demonstrate that 4RC consistently outperforms prior methods across a wide range of challenging 4D reconstruction benchmarks. Looking ahead, unified models such as 4RC, which jointly reason about geometry and motion, represent a promising direction toward more general-purpose perceptual systems.

## Acknowledgements

This research is supported by cash and in-kind funding from NTU S-Lab and industry partner(s).

## Impact Statement

This paper presents work whose goal is to advance the field of machine learning, with a particular focus on 4D reconstruction. The proposed approach has the potential to benefit applications in robotics, augmented/virtual reality, and content creation. While the method may have potential societal consequences, we do not identify any that require specific highlighting here.

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

# Appendix

## A. Additional Implementation Details

### A.1. Architecture Details

We adopt the ViT-Giant (ViT-G) architecture from DINOv2 (Oquab et al., 2023) as our encoder, which consists of 40 transformer layers with a feature dimension of 1,536 and employs 24 attention heads. The encoder weight is initialized from Depth Anything 3 (DA3) (Lin et al., 2026). For the geometry head, we follow a dual-DPT (Ranftl et al., 2021; Lin et al., 2026) design equipped with a lightweight MLP as the camera head. For the motion head, we employ a transformer-based decoder consisting of 4 layers of alternating self- and cross-attention with a hidden dimension of 1,536 and 16 attention heads. To generate high-resolution dense motion outputs, we leverage a DPT (Ranftl et al., 2021) upsampling strategy where we extract the feature tokens from the 19-th, 27-th, 33-rd, and 39-th blocks of the encoder. We therefore apply the motion head to these layers, concatenate the resulting outputs, and fuse them through the DPT head to regress the final dense motion displacement field.

### A.2. Dataset Details

We train 4RC on 7 datasets covering both dynamic and static environments. Table 5 details the statistics and sampling ratio of each dataset during training. For 3D motion learning, we leverage four dynamic datasets with ground-truth motion: PointOdyssey (Zheng et al., 2023), Dynamic Replica (Karaev et al., 2023b), Waymo (Sun et al., 2020), and Kubric (Greff et al., 2022). The motion supervision in these datasets varies from dense motion to sparse trajectories. Specifically for Kubric, we curate two subsets: 4,000 clips from the MOVi-F release (24 frames each) with dense motion annotations, and 6,000 clips from the CoTracker3 (Karaev et al., 2024) rendered training set (120 frames each) with sparse trajectory annotations. To ensure high-quality geometric reconstruction on static backgrounds, we additionally include three static datasets: DL3DV (Ling et al., 2024), ScanNet++ (Yeshwanth et al., 2023), and MVS-Synth (Huang et al., 2018).

*Table 5.* **Training dataset statistics.** We train 4RC on a mixture of 7 datasets. The motion annotation varies between dense maps and sparse trajectories depending on the dataset source. Static datasets naturally provide motion annotations, i.e., zero movement.

| Index | Dataset | Scene Type | Real / Synthetic | Dynamic / Static | Motion Annotation | Sampling (%) |
|---|---|---|---|---|---|---|
| 1 | PointOdyssey (Zheng et al., 2023) | Mixed | Synthetic | Dynamic | Sparse | 22.12 |
| 2 | Dynamic Replica (Karaev et al., 2023b) | Mixed | Synthetic | Dynamic | Sparse | 29.20 |
| 3 | Waymo (Sun et al., 2020) | Outdoor | Real | Dynamic | Dense | 4.42 |
| 4 | Kubric (Greff et al., 2022) | Object | Synthetic | Dynamic | Dense & Sparse | 26.55 |
| 5 | DL3DV (Ling et al., 2024) | Mixed | Real | Static | Dense | 8.85 |
| 6 | ScanNet++ (Yeshwanth et al., 2023) | Indoor | Real | Static | Dense | 3.54 |
| 7 | MVS-Synth (Huang et al., 2018) | Outdoor | Synthetic | Static | Dense | 5.31 |

### A.3. Training Details

During training, we apply standard data augmentations, including Gaussian blur ($p = 0.2$), ColorJitter ($p = 0.1$), and RandomGrayscale ($p = 0.05$). Video frames are sampled in strict temporal order with a random interval ranging from 1 to 5 frames. For motion supervision, we adopt a probabilistic sampling strategy. Specifically, in 20% of the training iterations, we supervise the model using all available motion ground truth. In the remaining 80%, we employ sparse supervision by retaining only the top 20–30% of points with the largest displacement magnitudes. Empirically, we find that this strategy filters out static or low-motion regions, prevents the dominance of zero-motion signals and accelerates convergence. For the ray map loss $\mathcal{L}_{\text{ray}}$ and the camera parameter loss $\mathcal{L}_{\text{cam}}$ in Equation 6, we adopt the loss formulation from DA3 (Lin et al., 2026) for supervision.

## B. Additional Experiments and Results

### B.1. Streaming Version of 4RC

To support causal and online 4D reconstruction, we further introduce a streaming variant of 4RC (S-4RC) which builds upon the STream3R (Lan et al., 2026) architecture. STream3R adapts a VGGT-style backbone into a transformer with causal

attention and serves as the encoder. Instead of performing bidirectional attention over the entire sequence, or interacting with a learnable state as in RNN-style designs, the encoder performs causal attention over cached tokens from previous frames.

Specifically, we replace our encoder with the pretrained STream3R backbone, which enforces unidirectional causal attention. We add a motion head to the model and fine-tune it using the proposed 4RC training objectives. Unlike standard 4RC, which processes the entire video in an offline manner, S-4RC operates sequentially and achieves per-frame latency. We cache the 4D latent representation $\mathcal{F}$ for all processed frames. This enables flexible motion queries from the current view to any past timestamp, as well as point tracking from past views to the current time. As shown in Table 6 and Figure 6, S-4RC achieves competitive performance in 4D reconstruction while operating in an online manner, without access to global temporal context. We also evaluate S-4RC on video depth estimation and camera pose estimation, as shown in Table 7 (a) and Table 7 (b). Note that S-4RC is trained for 20 epochs on 8 A100 GPUs.

*Table 6.* **4D reconstruction evaluation on tracking for S-4RC.** S-4RC enables online and streaming 4D reconstruction and achieves competitive performance compared to 4RC, even without access to global temporal context.

| Method | Point Tracking | | | | | | | | Dense Tracking | | | |
| | PO | | DR | | ADT | | PStudio | | Kubric | | Waymo | |
| | APD ↑ | EPE ↓ | APD ↑ | EPE ↓ | APD ↑ | EPE ↓ | APD ↑ | EPE ↓ | APD ↑ | EPE ↓ | APD ↑ | EPE ↓ |
|---|---|---|---|---|---|---|---|---|---|---|---|---|
| S-4RC | 73.29 | 0.3863 | 83.47 | 0.1970 | 86.12 | 0.1674 | 83.81 | 0.1795 | 75.60 | 1.168 | 46.02 | 1.971 |
| 4RC | 85.86 | 0.2498 | 88.65 | 0.1484 | 87.82 | 0.1480 | 87.32 | 0.1304 | 85.44 | 1.022 | 56.63 | 1.611 |

*Table 7.* **Video depth evaluation and camera pose estimation for S-4RC.**

| Method | Alignment | Bonn | | Sintel | |
| | | Rel ↓ | $\delta < 1.25$ ↑ | Rel ↓ | $\delta < 1.25$ ↑ |
|---|---|---|---|---|---|
| S-4RC | Scale | 0.074 | 97.4 | 0.311 | 63.9 |
| 4RC | Scale | 0.051 | 97.4 | 0.311 | 62.2 |
| S-4RC | Scale & Shift | 0.064 | 97.6 | 0.241 | 66.8 |
| 4RC | Scale & Shift | 0.048 | 97.3 | 0.249 | 67.0 |

*(a)* Video depth evaluation.

| Method | TUM | | | ScanNet | | |
| | ATE ↓ | RPE$_t$ ↓ | RPE$_r$ ↓ | ATE ↓ | RPE$_t$ ↓ | RPE$_r$ ↓ |
|---|---|---|---|---|---|---|
| S-4RC | 0.025 | 0.015 | 0.346 | 0.057 | 0.021 | 0.575 |
| 4RC | 0.010 | 0.008 | 0.314 | 0.032 | 0.012 | 0.437 |

*(b)* Camera pose estimation.

### B.2. Comparison with DA3 and Scale-up Training with More Reconstruction Data

We compare 4RC with DA3 (Lin et al., 2026) on video depth estimation, multi-view 3D reconstruction, and camera pose estimation. As shown in Table 8, 4RC achieves comparable performance to DA3 on multi-view 3D reconstruction and camera pose estimation, while being weaker on video depth estimation. This is expected since 4RC is primarily designed for 4D reconstruction and tracking, and the original model is trained on only 7 datasets, many of which are selected for tracking supervision rather than purely for 3D reconstruction. In contrast, DA3 is trained on 22 datasets. To further study this gap, we additionally collect more reconstruction datasets, including HyperSim (Roberts et al., 2021), TartanAir (Wang et al., 2020), MapFree (Arnold et al., 2022), MegaDepth (Li & Snavely, 2018), WildRGBD (Xia et al., 2024), ARKitScenes (Baruch et al., 2022), OmniWorld (Zhou et al., 2026), VirtualKITTI2 (Cabon et al., 2020), Unreal4K (Tosi et al., 2021), and Spring (Mehl et al., 2023). Together with the original datasets used in Table 5, this results in 17 training datasets in total. We fine-tune 4RC on this expanded dataset mixture for 20 epochs using 16 A100 GPUs.

The results show that additional reconstruction data consistently improves 3D-related performance, especially on video depth and multi-view reconstruction, while preserving the tracking ability of 4RC. As shown in Table 8 (a)-(c), 4RC trained with more datasets improves over the original model on most reconstruction metrics and outperforms DA3 on several multi-view reconstruction and pose estimation metrics. Meanwhile, Table 8 (d) shows that tracking performance remains stable, indicating that the added reconstruction data does not degrade the core 4D tracking capability.

*Table 8.* **Video depth, 3D reconstruction, camera pose, and point tracking evaluation for 4RC compared with DA3.**

| Method | Bonn | | Sintel | |
|---|---|---|---|---|
| | Rel ↓ | $\delta < 1.25$ ↑ | Rel ↓ | $\delta < 1.25$ ↑ |
| DA3 | 0.047 | 97.4 | 0.276 | 63.2 |
| 4RC (Ours) | 0.051 | 97.4 | 0.311 | 62.2 |
| 4RC (More datasets) | 0.049 | 97.3 | 0.239 | 68.3 |

*(a)* Video depth evaluation.

| Method | NRGBD | | | 7-Scenes | | |
|---|---|---|---|---|---|---|
| | Acc ↓ | Comp ↓ | NC ↑ | Acc ↓ | Comp ↓ | NC ↑ |
| DA3 | 0.031 | 0.032 | 0.925 | 0.050 | 0.046 | 0.790 |
| 4RC (Ours) | 0.036 | 0.034 | 0.912 | 0.034 | 0.051 | 0.783 |
| 4RC (More datasets) | 0.024 | 0.030 | 0.927 | 0.026 | 0.042 | 0.797 |

*(b)* Multi-view 3D reconstruction evaluation.

| Method | TUM | | | ScanNet | | |
|---|---|---|---|---|---|---|
| | ATE ↓ | $RPE_t$ ↓ | $RPE_r$ ↓ | ATE ↓ | $RPE_t$ ↓ | $RPE_r$ ↓ |
| DA3 | 0.012 | 0.010 | 0.311 | 0.035 | 0.015 | 0.650 |
| 4RC (Ours) | 0.011 | 0.009 | 0.315 | 0.032 | 0.013 | 0.437 |
| 4RC (More datasets) | 0.011 | 0.010 | 0.321 | 0.032 | 0.012 | 0.377 |

*(c)* Camera pose estimation.

| Method | PO | | DR | | ADT | | PStudio | |
|---|---|---|---|---|---|---|---|---|
| | APD ↑ | EPE ↓ | APD ↑ | EPE ↓ | APD ↑ | EPE ↓ | APD ↑ | EPE ↓ |
| 4RC (Ours) | 85.86 | 0.2498 | 88.65 | 0.1484 | 87.82 | 0.1480 | 87.32 | 0.1304 |
| 4RC (More datasets) | 85.78 | 0.2483 | 88.45 | 0.1503 | 88.09 | 0.1385 | 87.67 | 0.1402 |

*(d)* Point tracking evaluation.

## B.3. Additional Quantitative Evaluation on 4D Reconstruction

As a complement to the evaluation in Table 1, following WorldTrack (Feng et al., 2025) and TAPVid-3D (Koppula et al., 2024), we apply global median scale alignment to match the predicted points with the ground truth. This alignment is feasible since both the predictions and the ground-truth points are represented in a shared world coordinate system defined by the camera of the first frame. We additionally include a staged pipeline baseline composed of MonST3R (Zhang et al., 2025b) and SpatialTracker (Xiao et al., 2024). Comprehensive evaluations in Table 9 demonstrate that our method outperforms approaches specifically designed for point tracking as well as concurrent 4D reconstruction methods, achieving state-of-the-art results on 4 out of 6 datasets.

*Table 9.* **4D reconstruction evaluation on tracking under global median scale alignment.** As a complement to Table 1, we further evaluate our method on dense-view tracking (a) and sparse-view tracking (b) under *global median scale alignment* on dynamic datasets. Our method maintains strong performance across both evaluation protocols.

| Method | (a) Dense Tracking | | | | (b) Sparse Point Tracking | | | | | | | |
|---|---|---|---|---|---|---|---|---|---|---|---|---|
| | Kubric | | Waymo | | PO | | DR | | ADT | | PStudio | |
| | APD ↑ | EPE ↓ | APD ↑ | EPE ↓ | APD ↑ | EPE ↓ | APD ↑ | EPE ↓ | APD ↑ | EPE ↓ | APD ↑ | EPE ↓ |
| VGGT + CoTracker3 (Karaev et al., 2024) | - | - | - | - | 49.08 | 0.6532 | 74.73 | 0.2884 | 72.21 | 0.3548 | 66.28 | 0.3107 |
| MonST3R + SpatialTracker (Xiao et al., 2024) | - | - | - | - | 47.65 | 0.5917 | 55.49 | 0.8823 | 51.95 | 0.5362 | 50.16 | 0.4837 |
| SpatialTrackerV2 (Xiao et al., 2025) | - | - | - | - | 69.57 | 0.3780 | 73.43 | 0.2732 | 92.22 | 0.0915 | 74.16 | 0.2272 |
| St4RTrack (Feng et al., 2025) | 35.33 | 3.465 | 2.51 | 10.139 | 67.95 | 0.3140 | 73.74 | 0.2682 | 76.01 | 0.2680 | 69.67 | 0.2637 |
| TraceAnything (Liu et al., 2026a) | 27.37 | 1.952 | 2.06 | 12.564 | 39.83 | 1.0593 | 60.63 | 0.5758 | 75.65 | 0.2511 | 71.33 | 0.2727 |
| Any4D (Karhade et al., 2025) | - | - | - | - | 60.86 | 0.4194 | 68.39 | 0.3012 | 56.71 | 0.4320 | 60.03 | 0.3344 |
| V-DPM (Sucar et al., 2026) | 52.22 | 3.131 | 31.67 | 1.957 | 79.79 | 0.1994 | 76.38 | 0.2378 | 66.06 | 0.3426 | 76.36 | 0.1957 |
| **4RC (Ours)** | 55.38 | 1.525 | 39.55 | 1.864 | 80.27 | 0.2681 | 82.91 | 0.1889 | 84.28 | 0.1766 | 69.04 | 0.2603 |

## B.4. Additional Quantitative Evaluation on Depth Estimation

We additionally include the KITTI dataset (Geiger et al., 2013) and extend the video depth evaluation in the main paper. We compare with a broader set of baselines, including single-frame depth methods Marigold (Ke et al., 2024) and DepthAnything-V2 (Yang et al., 2024), video depth methods NVDS (Wang et al., 2023b), DepthCrafter (Hu et al., 2025), and ChronoDepth (Shao et al., 2024), and joint depth-and-pose estimation approaches Robust-CVD (Bârsan et al., 2018) and CausalSAM (Zhang et al., 2022). All results are aligned using per-sequence scale and shift, enabling a more comprehensive and fair comparison for video depth evaluation. As shown in Table 10, our method significantly outperforms existing depth estimation approaches and achieves competitive performance compared to the dynamic 3D reconstruction method Pi3 (Wang et al., 2025d). Notably, our method is not trained on large-scale 3D reconstruction datasets and is able to model dynamic

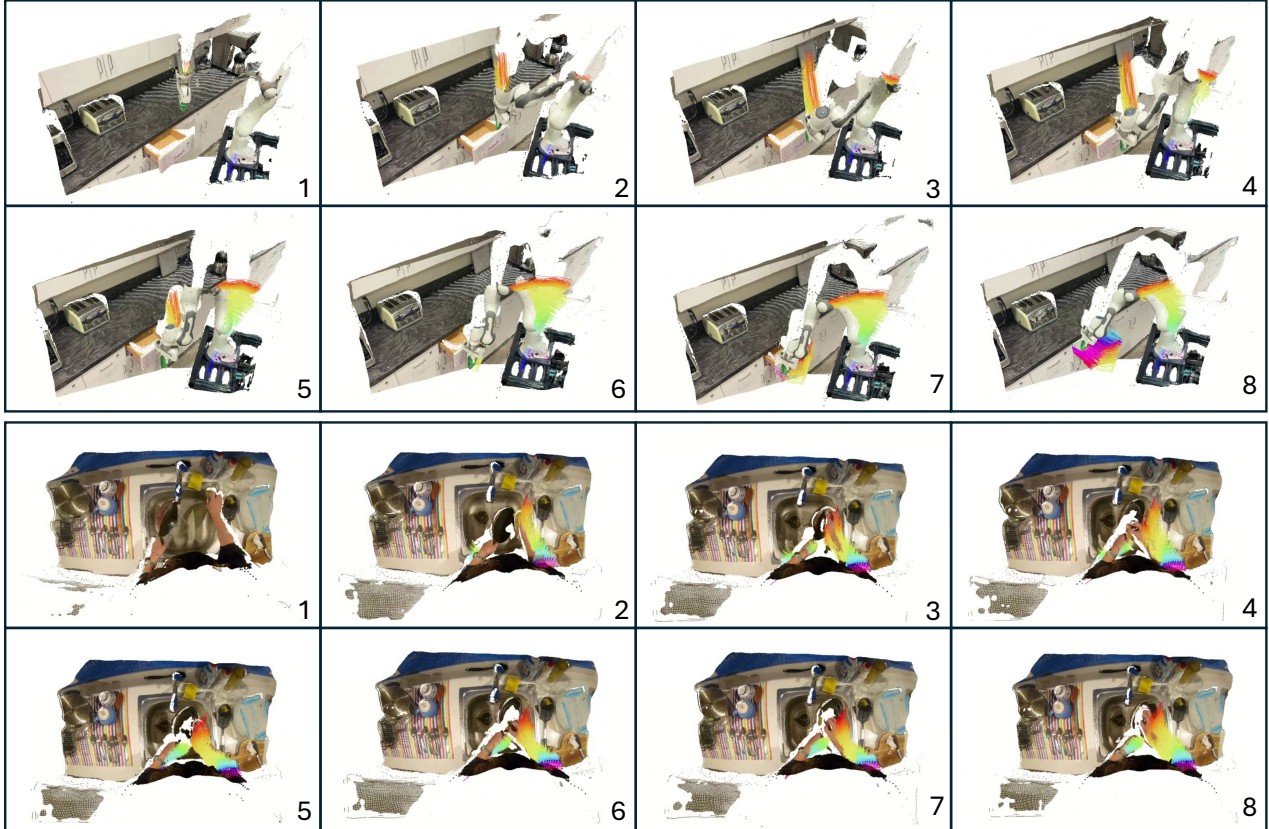

*Figure 6.* **The visualization of S-4RC results.** S-4RC can infer 3D geometry and motion in an online manner, which is beneficial for downstream tasks such as robotic motion planning and egocentric understanding.

object motion, rather than focusing solely on 3D reconstruction.

### B.5. More Visualizations

We further provide additional visualizations of our 4RC results, including camera poses, static reconstruction, dynamic reconstruction, and 3D tracking on in-the-wild videos in Figure 7.

### B.6. Video Demo

We also provide a demo video on our project page to showcase the qualitative 4D reconstruction results of 4RC and S-4RC.

## C. Limitations

While our method achieves unified and flexible feed-forward 4D reconstruction and shows stronger performance than concurrent 4D reconstruction methods, several limitations remain. First, our approach struggles in scenarios where geometric recovery is inherently difficult. These include regions with extreme depth (e.g., distant clouds), transparent objects, or floating artifacts where the base geometry lacks sharp depth boundaries. We expect that improved depth estimation methods (Xu et al., 2025) and future advances in 3D reconstruction will help alleviate these issues. Second, 4RC does not explore camera or depth priors as additional inputs. Recent works (Jang et al., 2025; Keetha et al., 2026; Liu et al., 2026b; Peng et al., 2026) suggest that incorporating reliable geometric priors can further improve reconstruction quality when accurate camera poses or depth cues are available. Third, we observe performance degradation in scenes with extreme or highly chaotic motion. This limitation mainly arises from the diversity of motion annotation in existing datasets, which provide insufficient supervision for such complex dynamics. Future work will explore scaling up training data to cover a broader range of motion patterns and kinematic diversity.

*Table 10.* **Video depth estimation on Bonn, Sintel, and KITTI datasets.** We compare a series of methods that explicitly predict video depth using per-sequence scale & shift alignment.

| Method | Bonn | | Sintel | | KITTI | |
|---|---|---|---|---|---|---|
| | Rel ↓ | $\delta < 1.25$ ↑ | Rel ↓ | $\delta < 1.25$ ↑ | Rel ↓ | $\delta < 1.25$ ↑ |
| Marigold (Ke et al., 2024) | 0.091 | 93.1 | 0.532 | 51.5 | 0.149 | 79.6 |
| Depth-Anything-V2 (Yang et al., 2024) | 0.106 | 92.1 | 0.367 | 55.4 | 0.140 | 80.4 |
| NVDS (Wang et al., 2023b) | 0.167 | 76.6 | 0.408 | 48.3 | 0.253 | 58.8 |
| ChronoDepth (Shao et al., 2024) | 0.100 | 91.1 | 0.687 | 48.6 | 0.167 | 75.9 |
| DepthCrafter (Hu et al., 2025) | 0.075 | 97.1 | 0.292 | 69.7 | 0.110 | 88.1 |
| Robust-CVD (Kopf et al., 2021) | - | - | 0.703 | 47.8 | - | - |
| CasualSAM (Zhang et al., 2022) | 0.169 | 73.7 | 0.387 | 54.7 | 0.246 | 62.2 |
| DUSt3R-GA (Wang et al., 2024b) | 0.156 | 83.1 | 0.531 | 51.2 | 0.135 | 81.8 |
| MASt3R-GA (Leroy et al., 2024) | 0.167 | 78.5 | 0.327 | 59.4 | 0.137 | 83.6 |
| MonST3R-GA (Zhang et al., 2025b) | 0.066 | 96.4 | 0.333 | 59.0 | 0.157 | 73.8 |
| Spann3R (Wang & Agapito, 2024) | 0.157 | 82.1 | 0.508 | 50.8 | 0.207 | 73.0 |
| CUT3R (Wang et al., 2025b) | 0.074 | 94.5 | 0.540 | 55.7 | 0.106 | 88.7 |
| VGGT (Wang et al., 2025a) | 0.049 | 97.2 | 0.202 | 72.7 | 0.057 | 96.6 |
| Pi3 (Wang et al., 2025d) | 0.044 | 97.5 | 0.229 | 73.2 | 0.038 | 98.4 |
| **4RC (Ours)** | 0.048 | 97.3 | 0.249 | 67.0 | 0.058 | 95.5 |

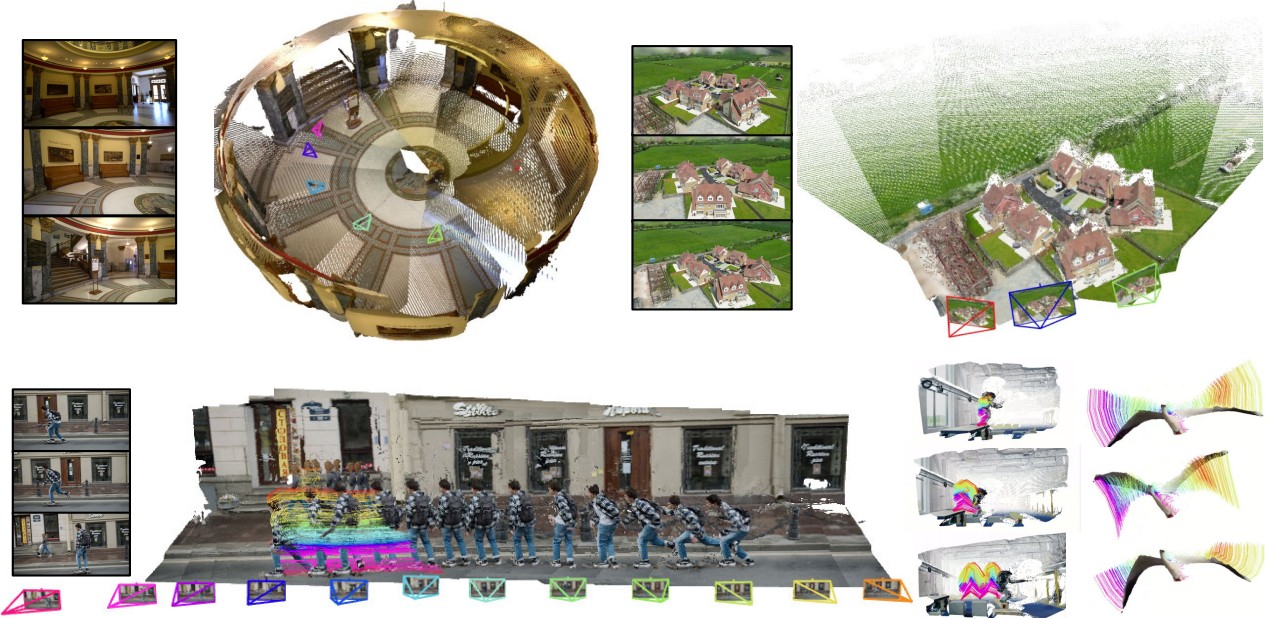

*Figure 7.* **Visualization using 4RC on in-the-wild videos** of camera poses, static reconstruction, dynamic reconstruction, and 3D tracking.

