# OpenReview forum: "4RC: 4D Reconstruction via Conditional Querying Anytime and Anywhere"
_ICML.cc/2026/Conference — ICML 2026 regular_

### Official Review · Reviewer_Fjyx · 2026-03-08

**Soundness:** 3
**Presentation:** 3
**Significance:** 3
**Originality:** 2
**Overall Recommendation:** 4
**Confidence:** 3

**Summary:**

This paper presents 4RC, a unified feed-forward framework for 4D reconstruction from monocular videos. The key idea is an "encode-once, query-anywhere/anytime" paradigm: a ViT backbone (initialized from DA3) encodes the video into a spatio-temporal latent, from which a conditional decoder queries geometry and motion at arbitrary timestamps. The 4D output is factorized into base geometry (depth, rays, camera params) and time-dependent 3D displacements (Eq. 2). Evaluation spans sparse/dense point tracking, camera pose estimation, multi-view reconstruction, and depth estimation, showing competitive results against concurrent methods.

**Compliance With Llm Reviewing Policy:**

Affirmed.

**Final Justification:**

The rebuttal fully resolved my concerns. I have no further questions, and I keep my score.

**Key Questions For Authors:**

1. Could you provide a direct comparison with DA3 (the initialization backbone) on depth and pose benchmarks, to disentangle how much of the performance comes from the pre-trained weights vs. the proposed 4RC modules?

2. What are the concrete runtime/accuracy trade-offs compared to D4RT when queried at comparable density? The current dismissal of per-point methods is not convincing without numbers.

3. Have you experimented with providing known camera poses or depth as additional input, as in MapAnything/WorldMirror-style conditioning?

**Limitations:**

yes

**Strengths And Weaknesses:**

**Strengths:**

S1. The unified encode-once paradigm is elegant and avoids the fragmented multi-stage pipelines of prior work (SpatialTracker, MonST3R). The factorized representation (base geometry + displacement) is well-motivated and validated in Table 4(b), where direct point prediction degrades significantly.

S2. Evaluation is thorough, covering tracking, pose, reconstruction, and depth across many datasets.

S3. Ablation studies (Table 4) are well-designed, systematically validating cross-attention, self-attention, and AdaLN in the motion head.

**Weaknesses:**

W1. The core novelty is somewhat incremental. The encoder is a standard ViT with alternating self-attention, initialized from DA3. The geometry head follows a dual-DPT design. The main new component is the lightweight motion decoder with AdaLN time conditioning—reasonable but not a major architectural advance.

W2. Tables 2 and 3 do not compare against DA3 as a standalone baseline for depth and camera poses. Since 4RC initializes from DA3 weights, it is hard to attribute improvements to the proposed components vs. the pre-trained backbone. This ablation is critical.

W3. The comparison with D4RT is only qualitative (Related Work - 4D Reconstruction). The paper claims dense querying is advantageous over D4RT's per-point approach, but a runtime/accuracy trade-off analysis is missing.

W4. Feed-forward 3D reconstruction works like MapAnything and WorldMirror are not cited or discussed. Whether providing known camera/depth priors as additional inputs could improve performance is unexplored.

W5. On Sintel depth (Table 3), 4RC achieves Rel=0.311 vs Pi3's 0.246 and VGGT's 0.297, showing a meaningful gap. The abstract's claim of outperforming prior methods "across a wide range" is overstated.

---

> ### Author Rebuttal · Authors · 2026-03-31
>
> We thank the reviewer for the constructive feedback. We appreciate the recognition of our `elegant encode-once paradigm`, `thorough evaluation`, and `well-designed ablations`. We address the concerns below.
>
> ---
>
> ### W1: Incremental architectural novelty
>
> Our method is specifically designed for 4D reconstruction. Our goal is not to discard the simple transformer backbone and strong geometric priors developed by recent feed-forward reconstruction methods, but to extend them into a unified 4D model. Rather than introducing additional architectural complexity, We view the main contribution of our work as the unified 4D formulation itself: an encode-once, query-anywhere and anytime paradigm, together with a factorized representation that decomposes base geometry and time-dependent relative motion. This design enables dense 4D querying from arbitrary source frames to arbitrary target timestamps, which is not supported by prior feed-forward 3D reconstruction methods.
>
> ---
>
> ### W2 & Q1 & W5: Comparisons with DA3 and improving 3D reconstruction
>
> Thank you for the question. We compare our method with DA3 on video depth evaluation, multi-view 3D reconstruction, and camera pose estimation in Tables A-C below. We find that our method is comparable to DA3 on multi-view 3D reconstruction and camera pose estimation, while being weaker on video depth evaluation. Since 4RC is designed for 4D reconstruction and tracking, and the original model is trained on only 7 datasets, many of which are chosen for tracking supervision rather than purely for 3D reconstruction. We note that DA3 is trained on 22 datasets, to further study this gap, we additionally collected more reconstruction datasets, including HyperSim, TartanAir, MapFree, MegaDepth, WildRGBD, ARKitScenes, OmniWorld, VirtualKITTI2, Unreal4K, and Spring. Together with the original datasets used in the paper, this gives a total of 17 datasets.
>
> We finetune on these datasets for 20 epochs with 16 A100 GPUs. The results show that 3D reconstruction performance (Tables A–C) further improves, while tracking performance (Table D) remains consistent, with no performance degradation. We will release the data processing scripts and open-source the checkpoint.
>
>
> ***Table A: Video depth evaluation compared with DA3.***
> | Method | **Bonn (Rel↓)** | **Bonn (δ<1.25↑)** | **Sintel (Rel↓)** | **Sintel (δ<1.25↑)** |
> | :--- | :---: | :---: | :---: | :---: |
> | **DA3**  | **0.047** | **97.4** | 0.276 | 63.2 |
> | **4RC (Ours)** | 0.051 | **97.4** | 0.311 | 62.2 |
> | **4RC (More datasets)** | 0.049 | 97.3 | **0.239** | **68.3** |
>
> ***Table B: Multi-View 3D reconstruction compared with DA3.***
> | Method | **NRGBD (Acc↓)** | **NRGBD (Comp↓)** | **NRGBD (NC↑)** | **7-Scenes (Acc↓)** | **7-Scenes (Comp↓)** | **7-Scenes (NC↑)** |
> | :--- | :---: | :---: | :---: | :---: | :---: | :---: |
> | **DA3** | 0.031 | 0.032 | 0.925 | 0.050 | 0.046 | 0.790 |
> | **4RC (Ours)** | 0.036 | 0.034 | 0.912 | 0.034 | 0.051 | 0.783 |
> | **4RC (More datasets)** | **0.024** | **0.030** | **0.927** | **0.026** | **0.042** | **0.797** |
>
> ***Table C: Camera pose estimation compared with DA3.***
> | Method | **TUM (ATE↓)** | **TUM (RPE_t↓)** | **TUM (RPE_r↓)** | **ScanNet (ATE↓)** | **ScanNet (RPE_t↓)** | **ScanNet (RPE_r↓)** |
> | :--- | :---: | :---: | :---: | :---: | :---: | :---: |
> | **DA3** | 0.012 | 0.010 | **0.311** | 0.035 | 0.015 | 0.650 |
> | **4RC (Ours)** | **0.011** | **0.009** | 0.315 | 0.032 | 0.013 | 0.437 |
> | **4RC (More datasets)** | 0.011 | 0.010 | 0.321 | **0.032** | **0.012** | **0.377** |
>
> ***Table D: Point tracking evaluation.***
> | Method | **PO (APD↑)** | **PO (EPE↓)** | **DR (APD↑)** | **DR (EPE↓)** | **ADT (APD↑)** | **ADT (EPE↓)** | **PStudio (APD↑)** | **PStudio (EPE↓)** |
> | :--- | :---: | :---: | :---: | :---: | :---: | :---: | :---: | :---: |
> | **4RC (Ours)** | **85.86** | 0.2498 | **88.65** | **0.1484** | 87.82 | 0.1480 | 87.32 | **0.1304** |
> | **4RC (More datasets)** | 85.78 | **0.2483** | 88.45 | 0.1503 | **88.09** | **0.1385** | **87.67** | 0.1402 |
>
> ---
>
> ### W3 & Q2: Comparisons with D4RT
>
> D4RT is concurrent work at the time of submission. Since its code is not yet publicly available, we are unable to conduct a direct quantitative comparison at this stage.
>
> ---
>
> ### W4 & Q3: Discussion on Mapanything and related work
>
> Thank you for the helpful suggestions. We will add and discuss the related work including Pow3R [Jang et al., 2025], MapAnything [Keetha et al., 2025], WorldMirror [Liu et al., 2025], and OmniVGGT [Peng et al., 2025] in the revised version. We also agree that exploring camera or depth priors as additional inputs is an interesting direction. Such priors may further improve reconstruction quality when reliable geometric cues are available. At the same time, this direction is largely orthogonal to the core contribution of our paper, which is to study whether a unified feed-forward model can directly learn 4D reconstruction and motion querying from monocular videos.

---

> > ### Author Rebuttal · Reviewer_Fjyx · 2026-04-02
> >
> > I have no further questions, and I keep my score.

---

> > > ### Author Response · Authors · 2026-04-02
> > >
> > > We sincerely thank the reviewer Fjyx for the positive evaluation and for recognizing that the concerns have been adequately addressed. We appreciate the reviewer’s time and consideration.

---

### Official Review · Reviewer_fvyd · 2026-03-10

**Soundness:** 3
**Presentation:** 3
**Significance:** 3
**Originality:** 3
**Overall Recommendation:** 4
**Confidence:** 4

**Summary:**

This paper presents a unified feed-forward transformer-based framework for 4D reconstruction from monocular videos. Unlike prior methods that decouple motion from geometry, rely on multi-stage processes, or produce limited outputs, 4RC jointly models dense 3D geometry and motion in a single pass. It enables flexible querying of 3D geometry and dense motion from any source frame to any target timestamp.

**Compliance With Llm Reviewing Policy:**

Affirmed.

**Final Justification:**

The rebuttal has fully addressed my concern. I will keep my assessment.

**Key Questions For Authors:**

1. The paper briefly mentions S-4RC for online reconstruction, but details on its architecture modifications and latency metrics are limited. Could you provide more quantitative results?
2. How does the framework perform for long sequence 4D reconstruction and tracking?

**Limitations:**

Yes.

**Strengths And Weaknesses:**

Strengths:
1. The paper's method and results are strong, because it creates a unified way to reconstruct 4D scenes from videos in one quick pass, outperforming other methods in tracking and geometry.
2. The paper is well-written and clearly presented.
3. The supplementary material also provides a streaming version of 4RC, which makes the method more practically valued.

Weaknesses:
1. The paper restricts the training sequence length to a range of [2, 18] views, which suggests that the model may struggle to process long video sequences.
2. The paper proposes decomposing 4D attributes into a "base geometry" and "time-dependent relative motion." While effective for general scenes, this assumption may break down in highly complex scenarios involving severe non-rigid deformations, fluid dynamics, or topological changes.

---

> ### Author Rebuttal · Authors · 2026-03-31
>
> We thank the reviewer for the positive assessment of our `method`, `presentation`, and `supplementary streaming extension`. We address the concerns below.
>
> ---
>
> ### W1: Training sequence length
>
> Similar to prior 3D methods (e.g., VGGT, DA3), we train on relatively short sequences (2-18 frames) for training efficiency. However, our method generalizes well to much longer videos during inference. For example, sparse tracking is evaluated on 64 frames, and video depth benchmarks such as Bonn include sequences exceeding 100 frames. As shown in Table D below, our method scales robustly from short clips to >100 frames.
>
> ---
>
> ### W2: Motion under complex dynamics
>
> As dicussed in the limitation section in Appendix C, we acknowledge that highly complex motion patterns remain challenging. We would like to emphasize that this limitation is shared by current feed-forward 4D reconstruction approaches. Notably, our motion factorization actually makes the learning for these cases easier rather than harder, since the model only needs to predict decomposed relative motion on top of the base geometry. We believe that with more diverse motion patterns in future datasets, this formulation could further improve its ability to model more challenging dynamics, including non-rigid deformations and topological changes.
>
> ---
>
> ### Q1: Details of S-4RC
>
> Thanks for the question. We provide additional implementation details, further quantitative evaluations (including video depth estimation and camera pose estimation), and efficiency analysis. We will include these discussions and results in the appendix of the revised version.
>
> **Implementation Details.** S-4RC is built on top of STream3R, which adapts a VGGT-style backbone into a transformer with causal attention and serves as the encoder. Specifically, instead of performing bidirectional attention over the entire sequence, or interacting with a learnable state as in RNN-style designs, the encoder performs causal attention efficiently over cached tokens from previous frames.
>
> **Further Evaluations.** Beyond the tracking evaluation for S-4RC, we further provide its quantitative results on video depth evaluation in Table A and camera pose estimation in Table B below.
>
> ***Table A: Video depth evaluation of S-4RC.***
> | Method | Alignment | **Bonn (Rel↓)** | **Bonn (δ<1.25↑)** | **Sintel (Rel↓)** | **Sintel (δ<1.25↑)** |
> | :--- | :---: | :---: | :---: | :---: | :---: |
> | **S-4RC** | Scale | 0.074 | 97.4 | 0.311 | 63.9 |
> | **4RC** | Scale | 0.051 | 97.4 | 0.311 | 62.2 |
> | **S-4RC** | Scale & Shift | 0.064 | 97.6 | 0.241 | 66.8 |
> | **4RC** | Scale & Shift | 0.048 | 97.3 | 0.249 | 67.0 |
>
> ***Table B: Camera pose esitination of S-4RC.***
> | Method | **TUM (ATE↓)** | **TUM (RPE_t↓)** | **TUM (RPE_r↓)** | **ScanNet (ATE↓)** | **ScanNet (RPE_t↓)** | **ScanNet (RPE_r↓)** |
> | :--- | :---: | :---: | :---: | :---: | :---: | :---: |
> | **S-4RC** | 0.025 | 0.015 | 0.346 | 0.057 | 0.021 | 0.575 |
> | **4RC**   | 0.010 | 0.008 | 0.314 | 0.032 | 0.012 | 0.437 |
>
>
> **Efficiency.** We report encoder latency and inference FPS on 64-frame inputs (504×378) for both S-4RC and 4RC in Table C. 4RC exhibits video-level latency (requiring all frames before output), whereas S-4RC operates with frame-level latency, producing outputs incrementally. We report the latency of the last frame and total FPS. When a new frame arrives, 4RC re-encodes all frames, while S-4RC processes only the incoming frame with KV cache, resulting in significantly lower last-frame latency.
>
> ***Table C: Latency of S-4RC.*** All results are measured on a single A100 GPU.
>
> | Method | **Latency type** | **Latency of last frame (s)** | **Average speed (fps)** |
> | :--- | :---: | :---: | :---: |
> | **S-4RC** | Frame | 0.07  | 17.32 |
> | **4RC** | video | 3.80 | 20.78 |
>
>
> ---
>
> ### Q2: Long-sequence 4D reconstruction and tracking
>
> In Table D, we further report point tracking performance on Dynamic Replica with varing sequence lengths of up to 112 frames. We find the 4RC remains stable from 32 to 112 frames. In contrast, S-4RC, which operates without global context and enables online processing, shows a slight performance drop as the sequence length increases. Besides, we have included qualitative tracking results on long sequences (~100 frames) in the demo video, further supporting our findings.
>
> ***Table D: Point tracking evaluation on Dynamic Replica with different numbers of frames.***
> | Num of frames | 32 |  | 64 | | 80 |  | 96 | | 112 | |
> | :--- | :--- | :---: | :--- | :---: | :--- | :---: | :--- | :---: | :--- | :---: |
> | **Method** | **APD↑** | **EPE↓** | **APD↑** | **EPE↓** | **APD↑** | **EPE↓** | **APD↑** | **EPE↓** | **APD↑** | **EPE↓** |
> | **4RC** | 88.44 | 0.1506 | 88.65 | 0.1484 | 88.52 | 0.1484 | 88.43 | 0.1493 | 88.30 | 0.1500 |
> | **S-4RC** | 84.20 | 0.1915 | 83.47 | 0.1970 | 83.36 | 0.1975 | 82.85 | 0.2019 | 82.74 | 0.2024 |

---

> > ### Author Rebuttal · Reviewer_fvyd · 2026-04-03
> >
> > Thanks for the detailed reply. I will keep my score.

---

> > > ### Author Response · Authors · 2026-04-04
> > >
> > > We sincerely thank the reviewer fvyd for the positive evaluation and for recognizing that the concerns have been adequately addressed. We appreciate the reviewer’s time and consideration.

---

### Official Review · Reviewer_KKkT · 2026-03-11

**Soundness:** 3
**Presentation:** 3
**Significance:** 2
**Originality:** 2
**Overall Recommendation:** 4
**Confidence:** 3

**Summary:**

This paper proposes 4RC, a unified feed-forward framework for 3D reconstruction and point tracking. Unlike previous methods that decouple motion and geometry understanding, 4RC learns a holistic 4D representation that jointly captures dense scene geometry and motion dynamics. The overall architecture of 4RC includes a ViT encoder and two separate geometry and motion heads. The paper conducts experiments on 3D/4D reconstruction tasks, and 4RC outperform other baselines.

**Compliance With Llm Reviewing Policy:**

Affirmed.

**Final Justification:**

My concerns have been solved. I will keep my score.

**Key Questions For Authors:**

Please consider replying to the weaknesses.

**Limitations:**

yes

**Strengths And Weaknesses:**

Strengths:

1. This paper gives an elegant formulation of the 4D reconstruction task in Eq. 2. This factorization is intuitive and helps to simplify the 4D reconstruction task.

2. The design of the motion head is interesting. It can flexibly query the 4D motion for arbitrary positions at arbitrary views and timestamps.

3. The paper evaluates the method across several benchmarks and tasks, and 4RC consistently outperforms existing methods.


Weaknesses:

1. My primary concern lies in the conceptual novelty of this paper. Despite the strong empirical evaluation, the overall architecture largely follow previous work. It looks like attaching an addition motion head to 3D reconstruction methods like DP3, VGGT, PI3. The motion head is also similar with Any4D. The authors said that "Any4D predicts motion only relative to the first frame", but given the camera poses are also predicted, people can easily transform the motion between different coordinates.

2. Based on Eq. 5, the motion head only relies on the query token and the target spatial tokens. In this case, the motion head cannot exploit the rich information from other intermediate frames, which is a large waste. Besides, the target tokens also restrict the interpolation, so the model cannot predict the point position in the continuous time space (e.g. between two frames).

3. The model conduct the 4D reconstruction in a shared world coordinate system. It is not clear how to define this world coordinate and whether the definition of the world coordinate would affect the performance.

---

> ### Author Rebuttal · Authors · 2026-03-31
>
> We thank the reviewer for recognizing our `elegant formulation` and the `strong empirical performance` across benchmarks. We address the concerns below.
>
> ---
>
> ### W1: Relation to prior 3D reconstruction methods and Any4D
>
> We agree that 4RC builds on recent advances in feed-forward 3D reconstruction. Our goal is not to discard these strong geometric priors, but to extend them into a unified 4D model via a factorized formulation. To this end, we introduce an *encode-once, query-anywhere and anytime* paradigm that learns a shared spatio-temporal latent and supports dense motion queries from an arbitrary source frame to an arbitrary target timestamp.
>
> This differs from prior methods in both scope and flexibility. Any4D predicts motion only from the first frame, and thus cannot create motion predictions for points or objects that are only visible in later frames; such missing motion information cannot be recovered through camera transformation. In contrast, our 4RC directly supports motion queries from any source frame, which is essential for achieving complete 4D reconstruction.
>
> ---
>
> ### W2: Use of global context and motion interpolation
>
> Thanks for the question. Our decoder is lightweight by design. Although the motion head operates only on the query and target tokens, these tokens are not limited to local information. The full video is first jointly encoded by the backbone Encoder into a unified 4D latent, so the query tokens and target tokens in Eq. (5) already contain global spatio-temporal context aggregated from the entire sequence. The motion head then performs query-specific retrieval from this globally contextualized latent, rather than re-encoding all frames for every query. Such joint encoding and lightweight decoding bring both efficiency and accuracy. We further finetune our model by allowing the motion head to cross-attend to all available intermediate frames' latents (i.e., $\hat{Z}_i$ for all target frames index $i \in \{1, \dots, N\}$). As shown in Table A below, we can see that this not only increases the time and peak GPU memory used for motion decoding, but even causes the accuracy to drop, as the context information contains many more redundant tokens.
>
> ***Table A: Dense track evaluation on Kubric and Waymo datasets, each containing 50 examples of 24 frames at a resolution of (504, 378).*** The results are tested on a single A100 GPU.
> | Method | **Kubric (APD↑)** | **Kubric (EPE↓)** | **Waymo (APD↑)** | **Waymo (EPE↓)** | **Motion Decode Speed (fps↑)** | **Motion Decode Memory (GB↓)** |
> | :--- | :---: | :---: | :---: | :---: | :---: | :---: |
> | **4RC (Attend all target tokens)** | 83.09 | 1.0631 | 55.78 | 1.629 | 11.73 | 10.47 |
> | **4RC (Ours)** | 85.44 | 1.022 | 56.63 | 1.611 | 47.43 | 3.93 |
>
> Regarding motion interpolation in continuous time, we agree this is an interesting direction. However, it is not the target setting of this study. Our method is designed for 4D reconstruction and tracking over the given input frames, which aligns with the standard setting in existing point-tracking and feed-forward 4D reconstruction benchmarks. Extending our framework to continuous-time prediction would be a meaningful direction for future work, but we consider it as complementary rather than essential to the current formulation.
>
> ---
>
> ### W3: Choice of world coordinate system
>
> As stated in Section 3.1, all point maps are represented in a world coordinate system defined by the camera of the first frame. This follows a common convention in prior works such as DUSt3R and its follow-ups (e.g., CUT3R, VGGT, DA3)
>
> We further study the impact of coordinate choices for motion in Section 4.4. Predicting motion directly in a shared world coordinate system performs better than predicting it in per-view local camera coordinates, suggesting that a shared global space is indeed more stable for smooth tracking. However, both are still worse than our final displacement-based formulation, which is why we adopt the factorized motion representation.
>
> ---
>
> We hope this clarifies the reviewer’s concern. If anything remains unclear, we would be happy to provide further clarification and discuss it in more detail.

---

> > ### Author Rebuttal · Reviewer_KKkT · 2026-04-03
> >
> > This rebuttal helps to solve my concerns. I will keep my score.

---

> > > ### Author Response · Authors · 2026-04-04
> > >
> > > We sincerely thank the reviewer KKkT for the positive evaluation and for recognizing that the concerns have been adequately addressed. We appreciate the reviewer’s time and consideration.

---

### Official Review · Reviewer_6bFu · 2026-03-11

**Soundness:** 3
**Presentation:** 3
**Significance:** 3
**Originality:** 3
**Overall Recommendation:** 4
**Confidence:** 4

**Summary:**

This work focuses on the topic of feed-forward 4D reconstruction, and proposes 4RC, a unified feed-forward framework for 4D reconstruction from monocular videos. Different from other feed-forward methods, this work introduces the paradigm of "encode-once, query-anywhere and anytime". Specifically, after achieving the encoded 4D latents with spatial-temporal informations, this work decodes the motion tracks using the proposed motion head, which uses the target time and the paired 4D latent of the source time and target time. Experimental results show that this work achieves comparable performances under different datasets of the 4D reconstruction task.

**Compliance With Llm Reviewing Policy:**

Affirmed.

**Final Justification:**

My concerns have been adequately addressed. I have no further questions, and I keep my score of 4.

**Key Questions For Authors:**

* Question about the efficiency of this paradigm. For example, if I want to obtain the tracking at all two points pairs in this video, could you please compare the efficiency of your proposed paradigm with other previous methods.

**Limitations:**

yes

**Strengths And Weaknesses:**

## Strength

* This work is well-written, and I can straightly understand the proposed point of this work.
* The proposed paradigm of "encode-once, query-anywhere and anytime" sounds good to me.
* Experimental results of ablations show the effectiveness of this work.

## Weakness

* This appears to be a typo: "(a) Geometry Head Architecture" in Table 4 should be "(a) Motion Head Architecture".
* Although I think the proposed paradigm of "encode-once, query-anywhere and anytime" sounds good to me, I still worry about the efficiency of this paradigm. Please see questions for more details.
* This work proposes a new paradigm, but the evaluation method is still the mainstream one. I think this may not be able to reflect the effectiveness of the method well. Would it be better to introduce more evaluation indicators? For example, analysis related to computing or time efficiency, analysis related to multiple queries.

---

> ### Author Rebuttal · Authors · 2026-03-31
>
> We thank the reviewer for the encouraging feedback, especially for noting the `clarity of the paper`, the `promise of the encode-once, query-anywhere and anytime paradigm`, and the `effectiveness of our ablation` studies.
>
> ---
>
> ### W1: Typos
>
> Thank you for pointing this out. We will correct the typos in the revised version.
>
> ### W2 & Q1: Efficiency analysis
>
> We provide an efficiency analysis under two settings. The first is the most common *1 to N* setting, where the motion of one query frame is predicted for all other target frames (i.e., long-term trajectory prediction). The second is the *N to N* setting as requested, where motion is queried for all source-target frame pairs.
>
> As shown in Table A, different methods make different trade-offs. St4RTrack, with engineering optimization through batching the input, can achieve high FPS, but this comes at the cost of GPU memory, and it remains a two-view method without global context. TraceAnything predicts Bézier curves to represent the whole trajectory for each view, which reduces inference time but provides limited geometry and motion accuracy. V-DPM predicts dynamic point maps for all target views jointly at a given query time, which limits its flexibility and also reduces inference efficiency in common tracking scenarios. Any4D can only predict motion from the first frame, so it is not included in this comparison. Overall, our method achieves a favorable balance between efficiency, flexibility, and memory usage. In particular, 4RC supports flexible dense motion querying from arbitrary source frames to arbitrary target timestamps while maintaining competitive runtime and substantially lower memory usage than methods that require jointly processing all frame pairs.
>
> ***Table A: Tracking efficiency comparisons with other methods.*** We test on N = 8 at the resolution nearest to (512, 384) based on each method's own patch size, and all results are measured on a single A100 GPU. We report FPS and peak GPU memory.
>
> | Method | **1 to N (fps↑)** | **1 to N (GB↓)** | **N to N (fps↑)** | **N to N (GB↓)** |
> | :--- | :---: | :---: | :---: | :---: |
> | **St4RTrack (batch)** | 25.76 | 5.57 | 3.54 | 29.38 |
> | **St4RTrack (sequential)**  | 10.72 | 2.60 | 1.36 | 2.60 |
> | **TraceAnything** | 11.89 | 6.28 | 11.89 | 6.28 |
> | **V-DPM** | 2.85 | 11.05 | 2.85 | 11.05 |
> | **4RC (Ours)** | 15.25 | 5.01 | 3.11 | 5.03 |
>
> ### W3: More evaluation indicators
>
> Thank you for the suggestion. In existing benchmarks of other 4D reconstruction methods, the evaluation mainly focuses on sparse point tracking from the first view. Since our method supports dense tracking from arbitrary query frames, we further construct a dense tracking setting from the middle view in Section 4.2 to better evaluate this capability.
>
> **Dense Tracking under Different Query Frames.** To further validate this flexibility, we additionally evaluate dense tracking on the Waymo dataset using different query indices, including the first frame (0), the middle frame (11), and the last frame (23). The results are shown in Table B below. Our method consistently outperforms other methods across different query views, which further supports the advantage of our query-based design.
>
> ***Table B: Dense tracking evaluation on Waymo with different query indices.***
>
> | Query Index | First(0) |  | Mid(11) | | Last(23) | |
> | :--- | :--- | :---: | :--- | :---: | :--- | :---: |
> | **Method** | **APD↑** | **EPE↓** | **APD↑** | **EPE↓** | **APD↑** | **EPE↓** |
> | **St4RTrack** | 18.92 | 6.110 | 19.98 | 6.359 | 18.60 | 6.804 |
> | **TraceAnything** | 22.59 | 4.641 | 21.25 | 4.313 | 25.87 | 4.046 |
> | **Any4D** | 20.38 | 4.270 | - | - | - | - |
> | **VDPM** | 39.20 | 2.449 | 41.44 | 1.948 | 42.13 | 2.237 |
> | **4RC (Ours)** | 55.18 | 1.999 | 56.63 | 1.611 | 54.68 | 2.074 |
>
> **Point Tracking under Different Sequence Lengths.** We further report point tracking performance under different numbers of input frames to evaluate robustness to sequence length. As shown in Table C, 4RC remains stable from 32 frames to 112 frames. In contrast, our streaming version, S-4RC, does not have access to full global context and instead performs online processing, so its performance drops slightly as the sequence length increases.
>
> ***Table C: Point tracking evaluation on Dynamic Replica with different numbers of frames.***
>
> | Num of frames | 32 |  | 64 |  | 80 |  | 96 |  | 112 |  |
> | :--- | :--- | :---: | :--- | :---: | :--- | :---: | :--- | :---: | :--- | :---: |
> | **Method** | **APD↑** | **EPE↓** | **APD↑** | **EPE↓** | **APD↑** | **EPE↓** | **APD↑** | **EPE↓** | **APD↑** | **EPE↓** |
> | **4RC** | 88.44 | 0.1506 | 88.65 | 0.1484 | 88.52 | 0.1484 | 88.43 | 0.1493 | 88.30 | 0.1500 |
> | **S-4RC** | 84.20 | 0.1915 | 83.47 | 0.1970 | 83.36 | 0.1975 | 82.85 | 0.2019 | 82.74 | 0.2024 |
>
> ---
>
> We hope this clarifies the reviewer’s concern. If anything remains unclear, we would be happy to provide further clarification and discuss it in more detail.

---

> > ### Author Rebuttal · Reviewer_6bFu · 2026-04-02
> >
> > I have no further questions, and I keep my score.

---

> > > ### Author Response · Authors · 2026-04-02
> > >
> > > We sincerely thank the reviewer 6bFu for the positive evaluation and for recognizing that the concerns have been adequately addressed. We appreciate the reviewer’s time and consideration.

---

### Decision · Program_Chairs · 2026-04-30

**Decision:**

Accept (regular)

**Comment:**

This paper receives initial scores of four weak accepts. After rebuttal, all the reviewers agree that the concerns have been resolved. Therefore, the AC recommends a decision of acceptance.